

# *Norrisanima miocaena*, a new generic name and redescription of a stem balaenopteroid mysticete (Mammalia, Cetacea) from the Miocene of California

Matthew S. Leslie[1,2,3], Carlos Mauricio Peredo[4] and Nicholas D. Pyenson[3,5]

[1] Biology Department, Swarthmore College, Swarthmore, PA, United States of America
[2] Department of Vertebrate Zoology, National Museum of Natural History, Smithsonian Institution, Washington, DC, United States of America
[3] Department of Paleobiology, National Museum of Natural History, Smithsonian Institution, Washington, DC, United States of America
[4] Department of Earth and Environmental Science, University of Michigan, Ann Arbor, MI, United States of America
[5] Department of Paleontology and Geology, Burke Museum of Natural History and Culture, Seattle, WA, United States of America

Corresponding author
Matthew S. Leslie,
matt.s.leslie@gmail.com,
mleslie2@swarthmore.edu

## ABSTRACT

Rorqual whales are among the most species rich group of baleen whales (or mysticetes) alive today, yet the monophyly of the traditional grouping (i.e., Balaenopteridae) remains unclear. Additionally, many fossil mysticetes putatively assigned to either Balaenopteridae or Balaenopteroidea may actually belong to stem lineages, although many of these fossil taxa suffer from inadequate descriptions of fragmentary skeletal material. Here we provide a redescription of the holotype of *Megaptera miocaena*, a fossil balaenopteroid from the Monterey Formation of California, which consists of a partial cranium, a fragment of the rostrum, a single vertebra, and both tympanoperiotics. *Kellogg (1922)* assigned the type specimen to the genus *Megaptera Gray (1846)*, on the basis of its broad similarities to distinctive traits in the cranium of extant humpback whales (*Megaptera novaeangliae* (*Borowski, 1781*)). Subsequent phylogenetic analyses have found these two species as sister taxa in morphological datasets alone; the most recent systematic analyses using both molecular and morphological data sets place *Megaptera miocaena* as a stem balaenopteroid unrelated to humpback whales. Here, we redescribe the type specimen of *Megaptera miocaena* in the context of other fossil balaenopteroids discovered nearly a century since Kellogg's original description and provide a morphological basis for discriminating it from *Megaptera novaeangliae*. We also provide a new generic name and recombine the taxon as *Norrisanima miocaena*, gen. nov., to reflect its phylogenetic position outside of crown Balaenopteroidea, unrelated to extant *Megaptera*. Lastly, we refine the stratigraphic age of *Norrisanima miocaena*, based on associated microfossils to a Tortonian age (7.6–7.3 Ma), which carries implications for understanding the origin of key features associated with feeding and body size evolution in this group of whales.

## INTRODUCTION

Rorqual whales include the largest vertebrates to have ever evolved in the history of life. Despite recent insights into evolutionary trends in body size for these taxa (*Slater, Goldbogen & Pyenson, 2017*), the overall phylogenetic relationships among extant lineages of rorquals remain a work in progress (*Árnason et al., 2018*). Specifically, questions remain regarding the monophyly of Balaenopteridae (relative to *Eschrichtius robustus* (*Lilljeborg, 1861*), or living gray whales) and the monophyly of the clade *Balaenoptera Lacépède, 1804* (with regard to the living genus *Megaptera Gray, 1846*).

Three main hypotheses for the relationships among *Eschrichtius*, *Balaenoptera*, and *Megaptera* reoccur predominantly in the recent literature (e.g., *Deméré et al., 2008*; *McGowen, Spaulding & Gatesy, 2009*; *Gatesy et al., 2013*; *Marx & Fordyce, 2015*; *Marx & Kohno, 2016*; *Slater, Goldbogen & Pyenson, 2017*; *Árnason et al., 2018*). The first is the traditional view of a monophyletic Eschrichtiidae and Balaenopteridae as sister clades with *Megaptera* sister to *Balaenoptera*, within Balaenopteridae (Fig. 1A). This view matches classification schemes built in the 20th century (e.g., *Rice, 1998*, and references therein) and the phylogenetic relationships derived from only morphological data sets (e.g., *Marx, 2011*; *Bosselaers & Post, 2010*; *Boessenecker & Fordyce, 2015*), with the exception of a single combined morphological and molecular data set (*Geisler et al., 2017*). The second reoccurring hypothesis (Fig. 1B) includes Eschrichtiidae nested within Balaenopteridae, and *Megaptera* within *Balaenoptera*. This overall pattern has been supported by molecular (*McGowen, Spaulding & Gatesy, 2009*; *Sasaki et al., 2006*) and combined morphological and molecular data sets (*Marx & Fordyce, 2015*; *Slater, Goldbogen & Pyenson, 2017*), including both fossil and extant taxa, as well from putative extinct members of Balaenopteridae and Eschrichtiidae *sensu lato* (*Marx & Kohno, 2016*; *Slater, Goldbogen & Pyenson, 2017*). By contrast, two molecular studies recovering the same pattern (*McGowen, Spaulding & Gatesy, 2009*; *Sasaki et al., 2006*) used only extant lineages, highlighting the inconsistent taxon sampling across these studies. The third hypothesis (Fig. 1C) built from combined morphological and molecular data, or strictly molecular data, places Eschrichtiidae and Balaenopteridae as sister clades with *Megaptera* nested within *Balaenoptera* and sister to *B. physalus Linnaeus, 1758* (*Sasaki et al., 2006*; *Deméré et al., 2008*; *Gatesy et al., 2013*). This hypothesis also recovers a monophyletic Balaenopteridae but does not recover a monophyletic genus *Balaenoptera* (Fig. 1C).

*Megaptera miocaena*, from the Monterey Formation of California, is a stem balaenopteroid described nearly a century ago (*Kellogg, 1922*). *Kellogg (1922)* originally assigned this species to the genus *Megaptera* based on the wide breadth of the cranium relative to its length and similarities to the extant *M. novaeangliae* (*Borowski, 1781*) in the tympanoperiotics. Two studies have discussed the taxonomic position of *Megaptera miocaena* in detail. *Deméré, Berta & McGowen (2005)* noted that *M. miocaena* lacks any of the autapomorphies of the extant *M. novaeangliae* and they explicitly opined that it is not a species of *Megaptera* and, therefore warrants placement in a new genus. Later, *Marx & Fordyce (2015)*, using phylogenetic analyses of morphology and DNA recovered *M. miocaena* as a stem taxon, outside of the group formed by living rorquals and gray whales.

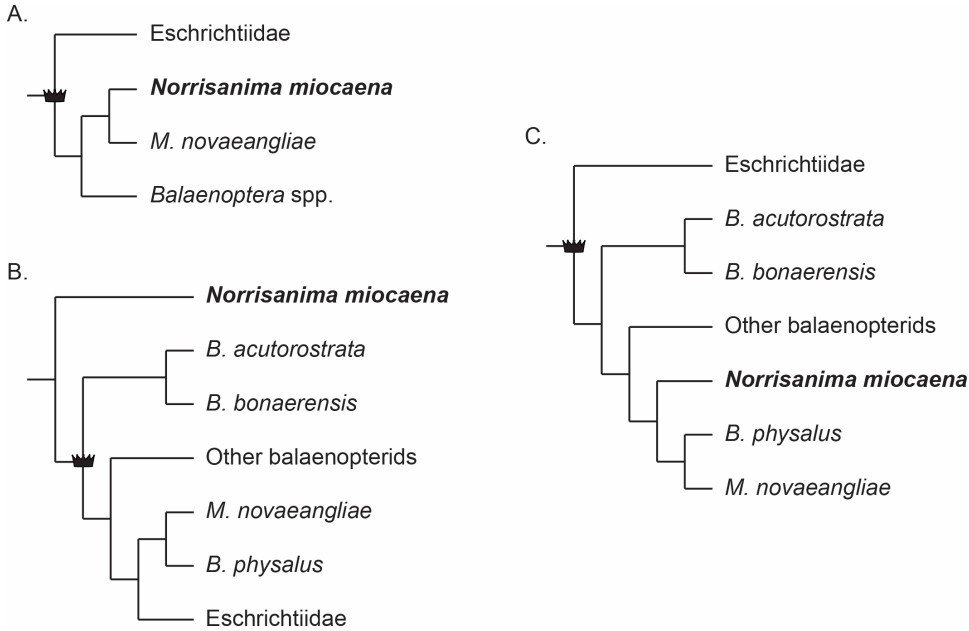

**Figure 1 Three frequently reoccurring phylogenetic hypotheses of the relationship between Balaenopteridae and Eschrichtiidae, as well as the relative placement of *N. miocaena*.** Crown symbol designates crown Balaenopteroidea. (A) The traditional view of monophyloetic Eschrichtiidae and Balaenopteridae (the latter including clades *Balaenoptera* and *Megaptera*). These relationships represent the historical view from the 20th century (e.g., *Rice, 1998*) as well as the results of most phylogenetic analyses based on morphological data (*Marx, 2011*; *Bosselaers & Post, 2010*; *Boessenecker & Fordyce, 2015*). (B) The hypothesized phylogenetic relationships supported by analyses of molecular (*McGowen, Spaulding & Gatesy, 2009*; *Sasaki et al., 2006*) as well as combined molecular + morphological data (*Marx & Fordyce, 2015*; *Slater, Goldbogen & Pyenson, 2017*). The latter studies including fossil and extant taxa as well from putative extinct members of Balaenopteridae and Eschrichtiidae *sensu lato* (*Marx & Kohno, 2016*; *Slater, Goldbogen & Pyenson, 2017*). (C) A third alternative hypothesis of large whale relationships resulting from molecular or combined molecular + morphological data that recovers a monophyletic Balaenopteridae, but does not recover a monophyletic genus *Balaenoptera* (*Sasaki et al., 2006*; *Deméré et al., 2008*; *Gatesy et al., 2013*). In this scenario, Eschrichtiidae and Balaenopteridae are sister clades with *Megaptera* nested within *Balaenoptera* and sister to *B. physalus*. Note the uncertainty regarding the placement of *N. miocaena* inside the balaenopterids, inside the clade *Megaptera*, and outside crown Balaenopteroidea (see text for further description of the data types and taxon sampling that support these hypotheses).

Several other recent phylogenetic analyses based on morphological and molecular data sets have also failed to recover the putative congeneric sister relationship between *M. miocaena* and *M. novaeangliae* (*Deméré et al., 2008*; *Gatesy et al., 2013*; *McGowen, Spaulding & Gatesy, 2009*; *Marx & Fordyce, 2015*; *Marx & Kohno, 2016*; *Slater, Goldbogen & Pyenson, 2017*). Interestingly, the only phylogenetic analyses that have recovered *M. miocaena* as the sister taxon of extant *M. novaeangliae* use exclusively morphological data (*Marx, 2011*; *Boessenecker & Fordyce, 2015*; *Boessenecker & Fordyce, 2017*).

Here we reexamine the holotype specimen of *M. miocaena*, provide a morphological basis for discriminating the putative similarities it shares with *M. novaeangliae*, and present the balaenopteroid synapomorphies that it lacks, affirming its status as a stem balaenopteroid. These morphological observations supplement the existing phylogenetic framework, using

comprehensive molecular and morphological datasets, which places *M. miocaena* outside of crown Balaenopteroidea (e.g., *Marx & Fordyce, 2015*; *Slater, Goldbogen & Pyenson, 2017*). Given this placement, we follow *Deméré, Berta & McGowen* (*2005*)'s recommendation to assign the specimen described by *Kellogg (1922)* to a new generic name: *Norrisanima*, nov. gen. Herein, we provide a detailed redescription of this taxon, explain how it differs from *Megaptera* and other crown and fossil balaenopteroids, and provide more details about its stratigraphic age and relevance for the evolution of rorquals and gray whales.

## METHODS

Anatomical terminology follows *Mead & Fordyce (2009)*. Permits for collection were not required, as the specimen was collected near Lompoc, California in 1919 and has been accessioned at the Smithsonian Institution ever since. For comparisons with crown balaenopteroids, we examined the following periotics in the collections of the Division of Mammals in the Department of Vertebrate Zoology at the Smithsonian's National Museum of Natural History (all right periotics except where noted): *M. novaeangliae* (USNM 486175), *Balaenoptera borealis Lesson, 1828* (USNM 504699), *B. physalus* (USNM 237566—left periotic), and *Balaenoptera bonaerensis Burmeister, 1868* (USNM 504953).

### 3D surface scanning—cranium

We used an Artec Eva (Artec Europe, Luxembourg) hand-held structured light scanner to create a 3D model of the cranium of the holotype specimen (USNM 10300). Because of the size and weight of the holotype specimen (>200 kg), we scanned the dorsal and ventral sides separately. We scanned at a rate of two frames per second and completed several scans to cover the surface of each side. All data cleaning, processing, and model creation were completed in the Artec Studio12 software package. We imported all scans for the dorsal side into a single project and performed a global registration, then aligned each scan incrementally using a set of three shared landmarks in the Align Tool Function and finished by cleaning and trimming the scan to remove data collected from the specimen housing jacket. The process was then repeated for the ventral side. Thus, scans from each side were aligned and trimmed in isolation from the other to create a composite model for each side. Then, we conducted a final global registration on the two models and again used the Align Tool to join the two halves into a single, final 3D model. We then completed another global registration and created a complete 3D model using the Fast Fusion tool; this model was not watertight. Holes in the model up to and including 150 pixels were filled using the Hole Fill tool; all other holes were left open. Most of the holes occurred in deep recesses where the scanner could not collect data, or where the storage jacket obscured both the dorsal and ventral sides of the cranium. Once the model was complete, we exported it as STL format for the distributed model (available in Supplemental Information), and for import into MeshLab (*Cignoni et al., 2008*) where we exported PNG image files. STL files are available at the Zenodo repository DOI: 10.5281/zenodo.3431395.

### CT scanning—tympanoperiotics

We scanned two periotics and one bulla from the holotype at the Smithsonian Institution Bio-Imaging Research Center in the Department of Anthropology at the National Museum of Natural History in Washington, D.C., USA. Computed tomography (CT) data were collected with a Siemens Somatom Emotion 6 (Siemens Medical Solutions, Erlangen, Germany) at slice thickness of 0.63 mm, resulting in a 3D reconstruction increment of 0.30 mm. DICOM files were processed by importing image files in Mimics Innovation Suite 19 (Materialise NV, Leuven, Belgium). In Mimics, we created a mask based on the threshold of bone relative to the nominal density of air. A 3D object was then created from this mask, and exported as a binary STL file. The STL file was then opened in MeshLab (*Cignoni et al., 2008*) for final editing to create 3D models and figures of the external morphology. The original DICOM files, STL files, and the 3D files of the cranium, are archived at Zenodo (http://zenodo.org) at the following DOI: 10.5281/zenodo.3431395.

### Phylogenetic nomenclature

We followed the recommendations of *Joyce, Parham & Gauthier (2004)* for the conversion of select ranked taxonomic cetacean names to phylogenetically defined ones in this study, following similar steps by *Pyenson et al. (2015)*. For these purposes, we used abbreviations NCN for New Clade Name and CCN for Converted Clade Name. Below, we clarify our precise definitions for these clades (see PhyloCode, 2014, Article 9.3; *Cantino & De Queiroz, 2014*), and we also provide full citations for the names of specifier species, when warranted.

### Nomenclatural acts

The electronic version of this article adheres to the amended International Code of Zoological Nomenclature (ICZN). Specifically, the new name contained in this work is available under the ICZN from the electronic version of this article. Both the nomenclatural acts and the published work itself are registered in ZooBank, the online registration system for the ICZN. The ZooBank Life Science Identifiers and the associated information can be viewed online by appending the LSID to the prefix "http://zoobank.org/" in any web-browser. The LSID for this publication is:

urn/lsid/zoobank.org/pub/95CFDD42-D8DB-4DC7-BFB3-5B34CCC6508C

## RESULTS

### Systematic paleontology

MAMMALIA *Linnaeus, 1758*
CETACEA *Brisson, 1762*
PELAGICETI *Uhen, 2008*
NEOCETI *Fordyce & De Muizon, 2001*
MYSTICETI *Gray, 1864*
PLICOGULAE *Geisler et al., 2011*
PAN BALAENOPTEROIDEA (NCN) (panstem-based version of Balaenopteroidea *Gray, 1868*)

*Norrisanima*, nov. gen., urn/lsid/zoobank.org/act/E777170E-03BC-40AA-A04B-65CE92C956BD

**Definitions:** 'Pan-Balaenopteroidea' refers to the panstem that includes crown Balaenopteroidea (CCN), and all other lineages closer to *Balaenoptera Lacépède, 1804* than to *Caperea Gray, 1864*, such as *Pelocetus calvertensis Kellogg, 1965*, *Norrisanima miocaena* (*Kellogg, 1922*) and *Parabalaenoptera bauliensis Zeigler, Chan & Barnes, 1997*. Crown group Balaenopteroidea refers to the crown clade arising from the last common ancestor of *Eschrichtius* and all named extant species of *Balaenoptera*. Given the potential paraphyly of both the family Balaenopteridae and the genus *Balaenoptera* (see 'Introduction'), we elect not to formalize crown concepts for these aforementioned taxonomic groups within Balaenopteroidea at this time.

**Type and only known species**: *Norrisanima miocaena* (*Kellogg, 1922*), new combination.

**Etymology**: Combining the surname Norris and the Latin *anima* (breath of life), the generic name honors the late Dr. Kenneth S. Norris and his son, Dr. Richard D. Norris, for their contributions to the natural history of California, marine mammalogy, and evolution in the marine realm. Aside from the holotype specimen's provenance from southern California, the epithet also honors the teaching and mentorship legacies of both Norrises at various campuses of the University of California, including the Scripps Institution of Oceanography, where RD Norris served on MSL's doctoral dissertation committee.

**Diagnosis:** Same as that of the only known species *Norrisanima miocaena*, new combination Figs. 2–8; Figs. S1–S6.

**Age:** Same as that of the only known species.

*Norrisanima miocaena*, new combination, LSID: urn:lsid:zoobank.org:act:E777170E-03BC-40AA-A04B-65CE92C956BD

**Diagnosis**: *N. miocaena* is a stem balaenopteroid that possesses the following autapomorphies: the lateral margins of the nasal are parallel; spreading of the anterolateral portion of the parietal on to the posteromedial corner of the supraorbital process of the frontal; the anteriormost point of the supraoccipital in dorsal view is in line with the anterior half or anterior edge of the supraorbital process; the zygomatic process of the squamosal is distinctly higher dorsoventrally than wide transversely; having a squared anterior apex of the supraoccipital shield; the tip of the postglenoid process pointing ventrally in lateral view, lacking a distinct ridge delimiting the insertion surface of the tensor tympani on medial side of the anterior process of the periotic; and having a superior process of the periotic present as a distinct crest forming the lateral wall of the suprameatal fossa.

**Holotype:** USNM 10300 is an incomplete cranium, including both tympanoperiotics as isolated material, an isolated lumbar vertebra, a disarticulated fragment of the vomer, and single non-phalanx fragment of the manus. *Kellogg (1922)* reported that the anterior portion of the rostrum and much of the right side of the skull were damaged or destroyed during excavation. In addition, the palatines were damaged during transport to the Smithsonian Institution. Additional plaster has been added since the original description—particularly to hold parts of the right side of the cranium (squamosal, including the zygomatic arch, and maxilla) together. The disarticulated vomer fragment has no patent connections with the cranium. The right tympanic bulla was also damaged, but most of the fragments were
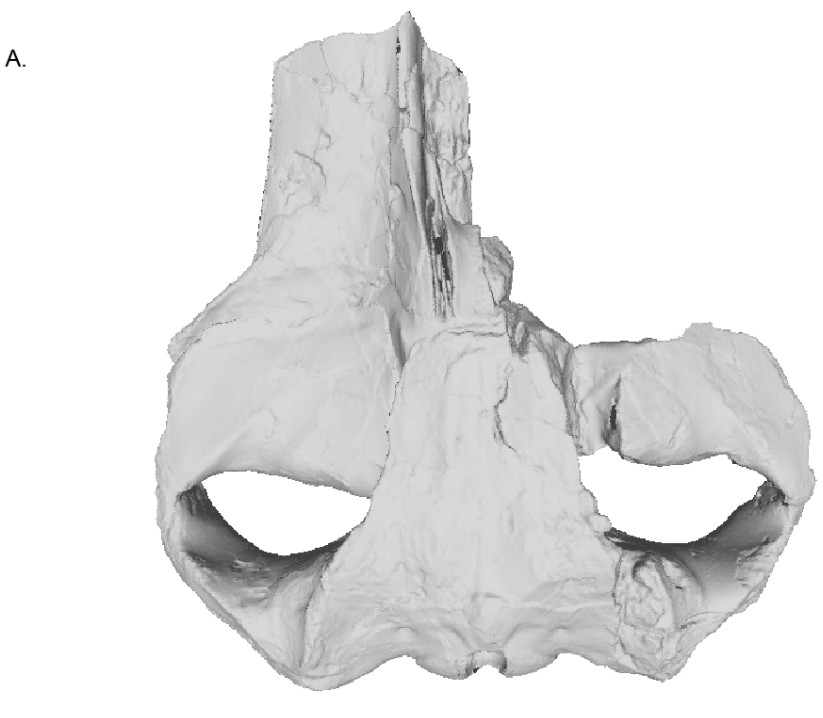

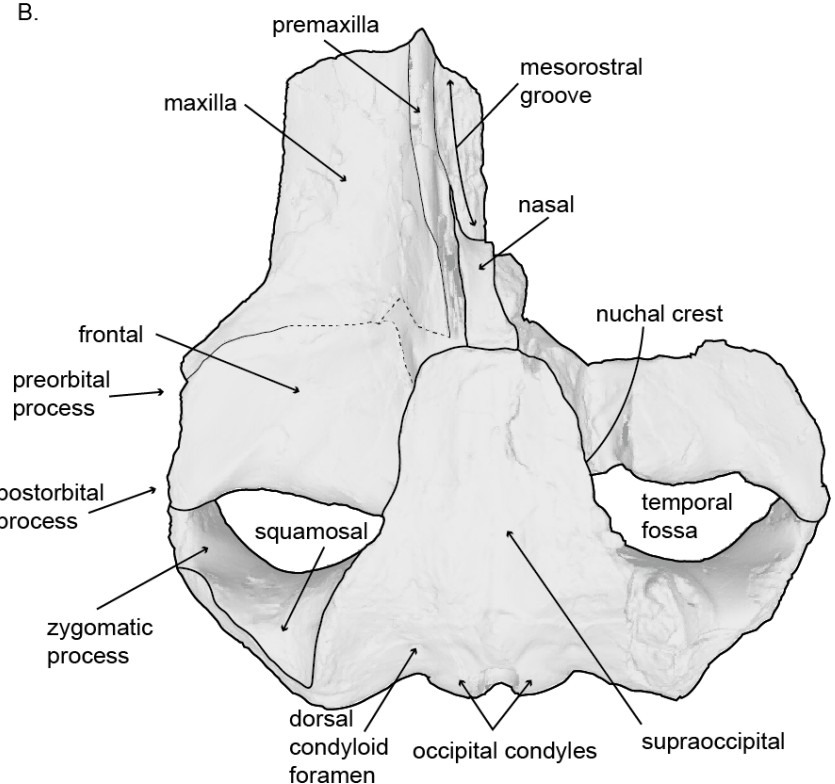

**Figure 2  Holotype skull (USNM 10300) of *Norrisanima miocaena* in dorsal view.** Dorsal view taken from 3D model created by structured light scanning. (B) Illustrated with a low opacity mask and interpretive line art. Dotted lines indicate uncertain boundaries.

A.

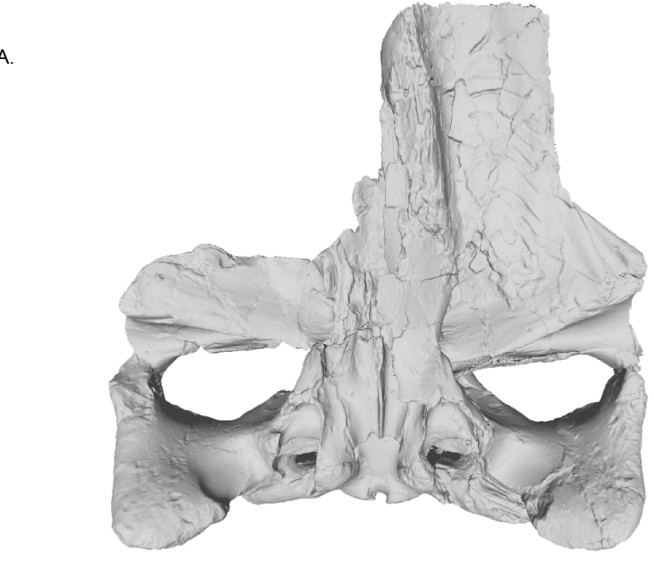

B.

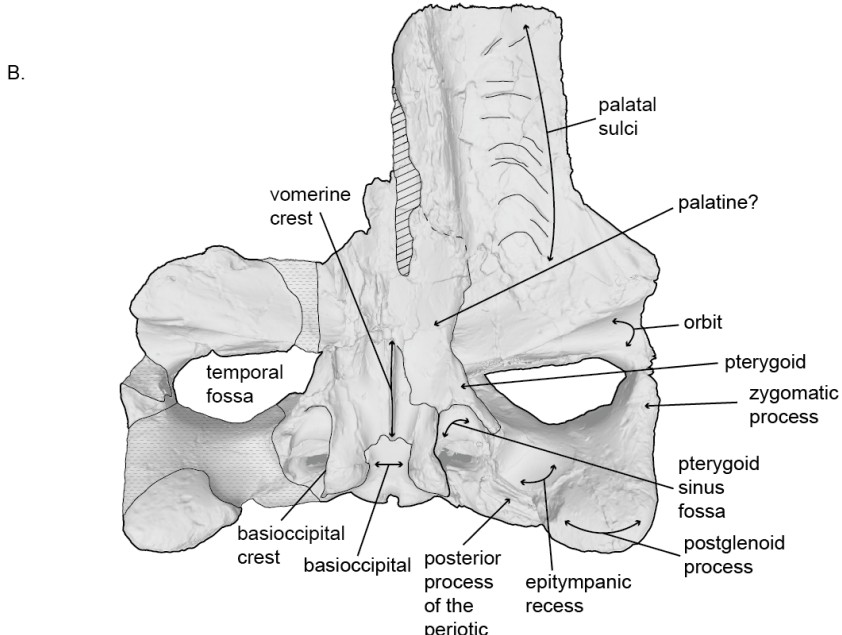

**Figure 3** **Holotype skull (USNM 10300) of** *Norrisanima miocaena* **in ventral view.** Ventral view taken from 3D model created by structured light scanning. B. Illustrated with a low opacity mask and interpretive line art. Stippling is matrix or plaster. Dotted lines indicate uncertain boundaries. Diagonal lines are broken or missing bone.

recovered and reconstructed. The right periotic is missing its posterior process but is otherwise complete.

**Type Locality:** The main Celite quarry of the Lompoc diatomite mines (after the Celite Corporation), also called the Johns-Manville quarry (after the Johns-Manville Company),

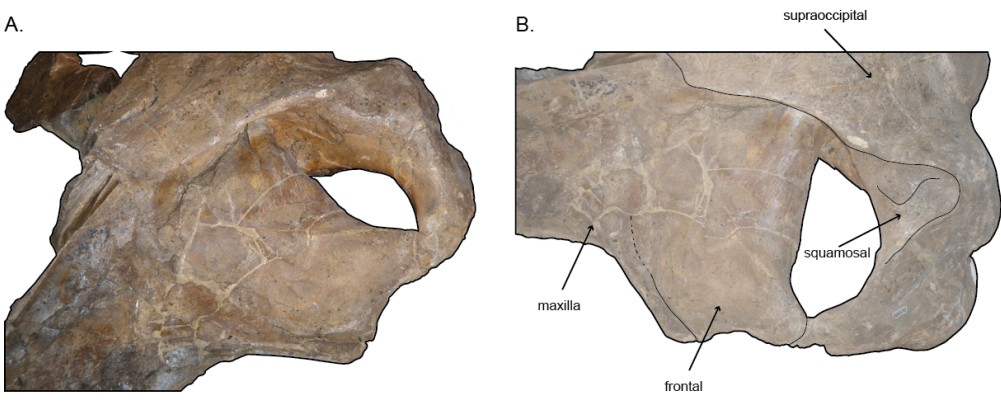

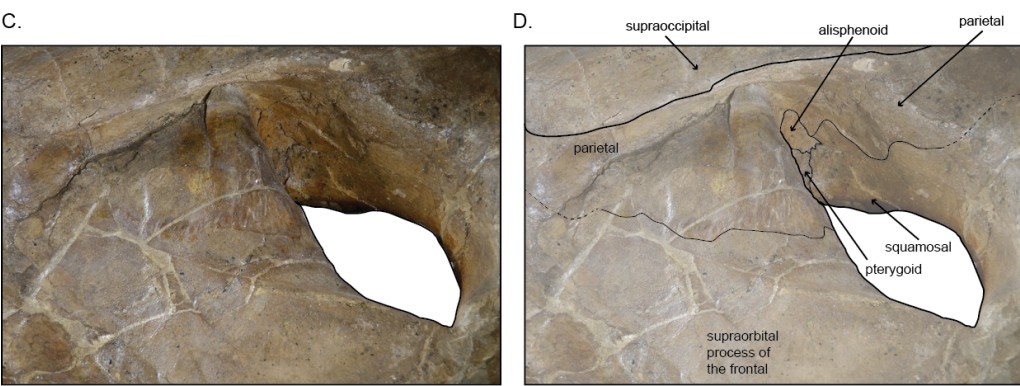

**Figure 4** **Left temporal wall and orbital tegion of holotype skull (USNM 10300) of *Norrisanima miocaena*.** (A) Anterodorsal; (B) dorsolateral; (C, D) dorsolateral enlarged. Dotted lines indicate uncertain boundaries.

near Lompoc, California, USA. *Kellogg (1922)* described the type locality as "one-half mile northwest of the northeast corner of township 6 north, range 34 west (Lompoc Quadrangle), on top of divide between drainage of San Miguelito Creek and Salsipuedes Creek, 3 miles south and east of Lompoc, Santa Barbara County, California". Using Google Earth and the current Lompoc Hills Quadrangle map (USGS NGA Ref No: X24K26325), which includes the township 6 borders, the approximate GPS coordinates for the excavation site are 34°36′56″N; 120°26′43″W with a margin of error of approximately 0.8 km. The Celite quarry mentioned for the type of *Norrisanima* is equivalent to the Johns-Manville quarry (see *Dibblee Jr, 1950*), but separate from the nearby Lakes Carbon Corporation quarry (which is the type locality of the fossil odobenid *Imagotaria downsi Mitchell, 1968*) and very likely separate from Celite Quarry No. 9, which is the type locality of the fossil crown otariid *Pithanotaria starri Kellogg, 1925*; see *Repenning & Tedford (1977)*.

**Formation:** Monterey Formation.

**Age:** Late Tortonian, between 7.6–7.3 Ma. *Kellogg (1922)* reported that the type specimen was discovered by quarrymen at the Celite Products Company (now Celite Corporation,

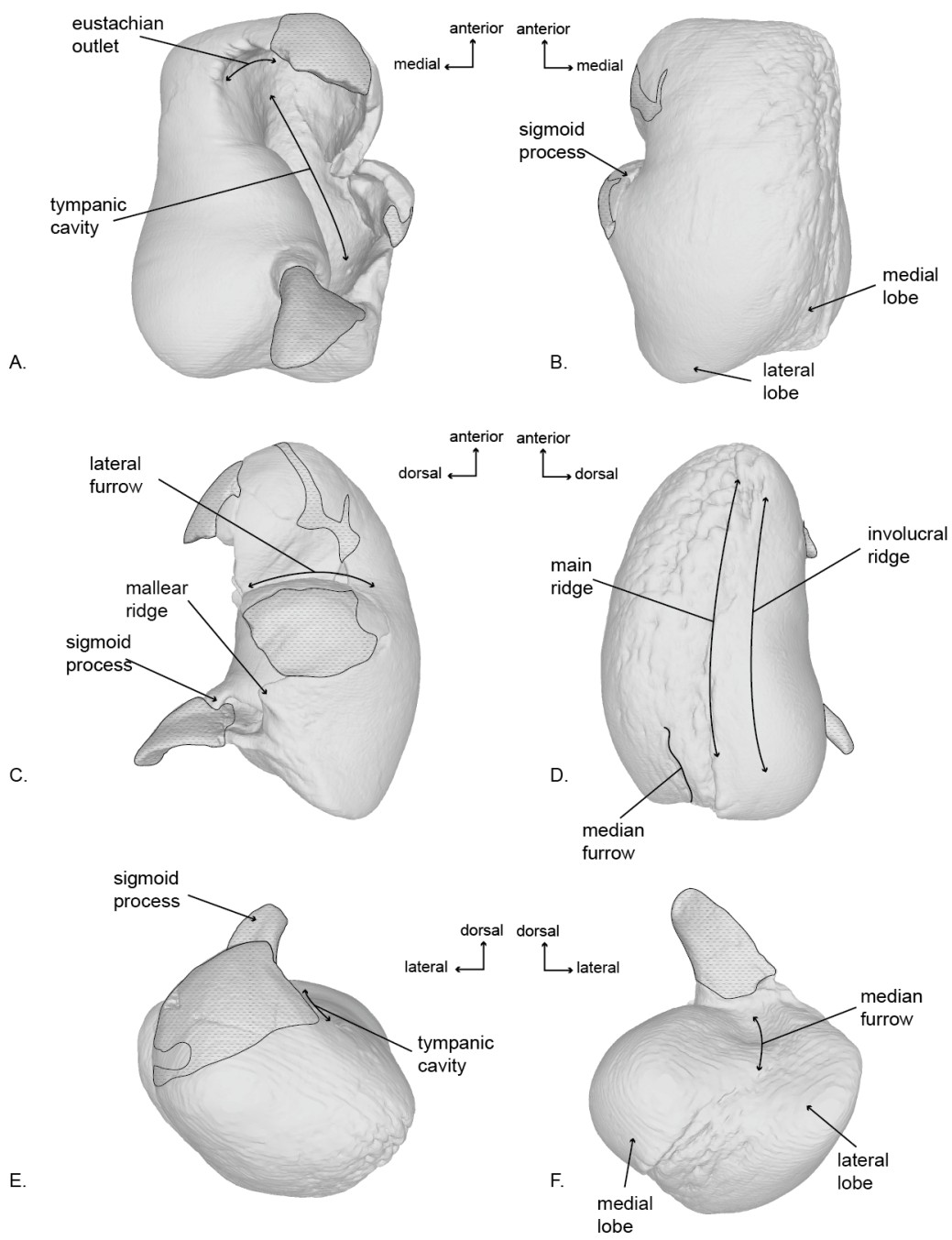

**Figure 5 Right tympanic bulla of the holotype (USNM 10300) of *Norrisanima miocaena*.** Image taken from 3D model created by CT scanning and illustrated with a low opacity mask and interpretive line art: (A) dorsal, (B) ventral, (C) lateral, (D) medial, (E) anterior, (F) posterior. Stippling is matrix or plaster.

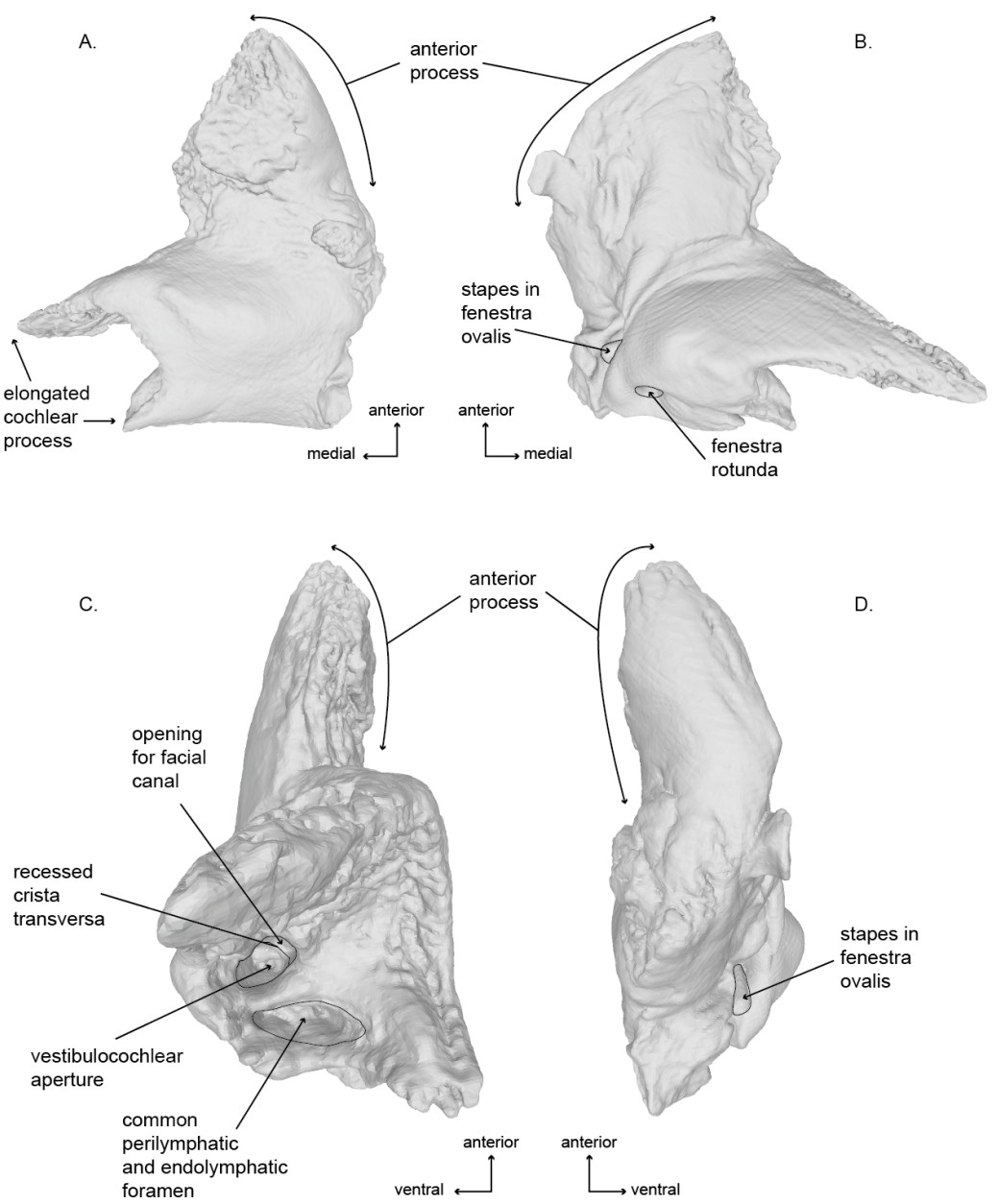

**Figure 6** **Right periotic of the holotype (USNM 10300) of *Norrisanima miocaena*..** Image taken from 3D model created by CT scanning: (A) dorsal, (B) ventral, (C) medial, (D) lateral.

or generally Celite) diatomite mines in Lompoc, California, in a horizon "about 150 ft below the quarry's surface" at the type locality. *Kellogg (1922)* noted, at the time, that a precise determination of the type specimen's stratigraphic origin would not be possible until the quarry deepened to that latter depth; to our knowledge, this determination never happened. At the time of *Kellogg* (*1922*)'s description, Kellogg correlated the Lompoc diatomite mines with the Temblor Formation (roughly middle to late Miocene), while work throughout the 20th century eventually assigned the Lompoc diatomites to the

A. *Caperea marginata*

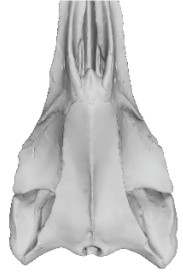

B. *Eubalaena australis*

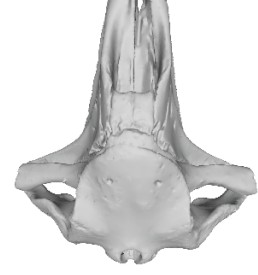

C. *Balaenoptera edeni*

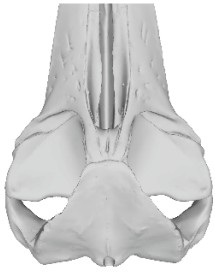

D. *Megaptera novaeangliae*

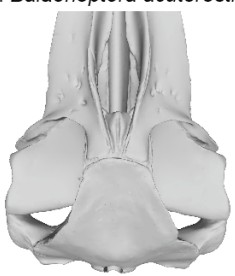

E. *Norrisanima miocaena*


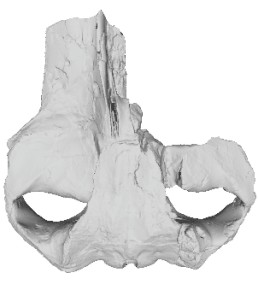

F. *Balaenoptera physalus*

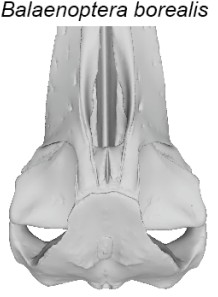

G. *Balaenoptera acutorostrata*

H. *Balaenoptera borealis*

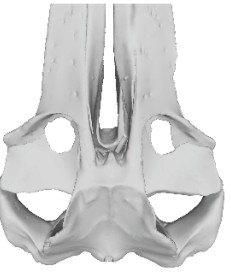

I. *Balaenoptera musculus*

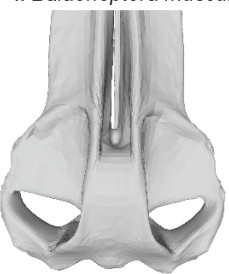

**Figure 7** **Comparisons of the vertex and dorsal surface of the cranium of *Norrisamina miocaena* with some extant baleen whale species based on available 3D models.** (A) *Caperea marginata Gray, 1846* (NHMUK 1876.2.16.1), (B) *Eubalaena australis* (NHMUK 1873.3.3.1), (C) *Balaenoptera edeni Anderson, 1878* (NHMUK 1920.12.31.1), (D) *Megaptera novaeangliae Borowski, 1781* (NHMUK 792a), (E) *Norrisanima miocaena* (USNM 10300), (F) *B. physalus* (NHMUK 1862.7.18.1), (G) *B. acutorostrata* (NHMUK 1965.11.2.1), (H) *B. borealis* (NHMUK 1934.5.25.1), (I) *B. musculus* (NHMUK 1892.3.1.1). All NHMUK scans were downloaded from https://doi.org/10.5519/0020467, with the exception of the blue whale which was acquired from https://sketchfab.com/NHM_Imaging (*Sabin et al., 2018*), and humpback whale, which was made available directly from the NHMH to the authors.

Monterey Formation. Relying on work by *Kleinpell (1938)*, *Barron (1986)* and *Behl & Ramirez (2000)* reported a late Miocene age for the Lompoc diatomites between 8.5 and 5.5 Ma (Late Tortonian to Messinian). Later, *Barron & Isaacs (2001)* revised the age of the Lompoc diatomites to 9.2–6.8 Ma, based on a detailed chronostratigraphic framework of the Monterey Formation. *Deméré, Berta & McGowen (2005)* argued that the Lompoc diatomites should be Tortonian in age (8.2–7.3 Ma), following *Chang & Grimm (1999)*. *Marx & Fordyce (2015)* cited *Chang & Grimm* (*1999*)'s upper age boundary of 7.3 Ma on the 218 m sequence of mineable diatomite exposed at the Celite quarry (reported to *Chang & Grimm (1999)* as "JA Barron, pers. comm., 1997") along with unpublished data from

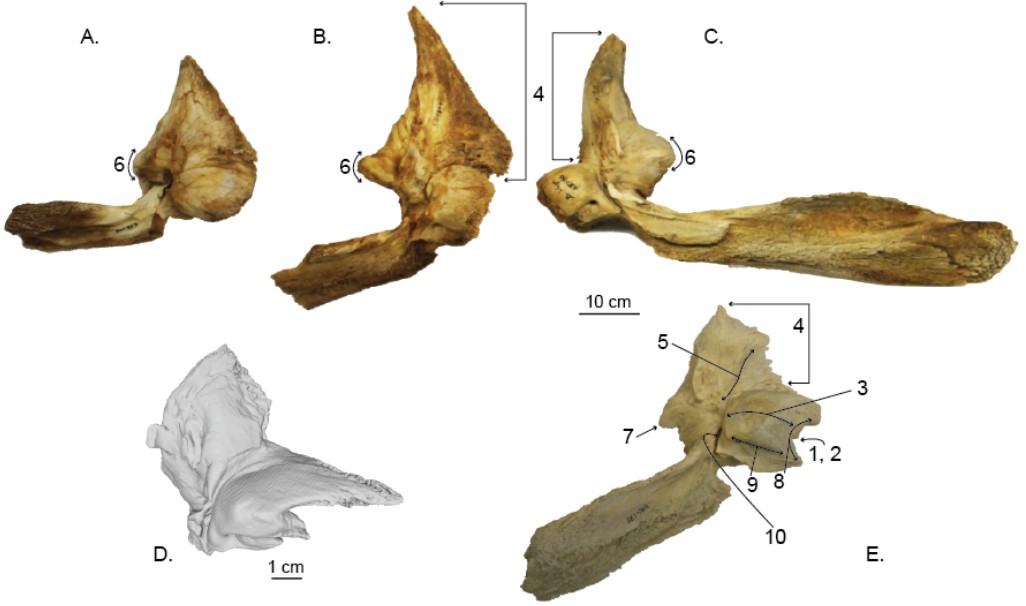

**Figure 8** Comparisons of the right periotic of *Norrisamina miocaena* with extant species within *Balaenoptera* and *Megaptera* (except for C, all specimens are the right periotic; all are shown in ventral view with the anterior oriented up). (A) *Balaenoptera bonaerensis* (USNM 504953), (B) *B. borealis* (USNM 504699) (C) *B. physalus* (USNM 237566), (D) *Norrisanima miocaena* (USNM 10300), (E) *Megaptera novaeangliae* (USNM 486175). *N. miocaena* is enlarged with a scale bar at 1 cm; the scale bar for other periotics is 10 cm. Numbered characters are listed in Table 1: (1) Aperture of cranial nerve (CN) ducts deeply recessed in pars cochlearis (PC), (2) Aperture of CN ducts erupt medial; not deflected dorsally, (3) Dorsoventrally flat and mediolaterally elongate PC, (4) Short and robust anterior process (AP) relative to size of PC, (5) Presence of lateral crest on the ventral surface of AP, (6) Sharply pointed triangular flange, (7) Posteriorly deflected triangular flange, (8) Concave medial margin of PC, (9) Transverse ridge on ventral surface of PC, (10) Deep invagination of fenestra ovalis.

N. Kohno (pers. comm. to *Marx & Fordyce (2015)*, in 2010), who assigned diatoms collected from the matrix surrounding the cranium of *N. miocaena* to the *Rouxia californica* diatom subzone (NPD7A) of *Akiba (1986)*. The NPD7A subzone now ranges across the current boundaries of the Tortonian and Messinian with an age range of 7.6–6.5 Ma (see *Barron, Lyle & Koizumi, 2001*), with the older age providing a lower age bound. Thus, *Marx & Fordyce (2015)* argued that the preponderance of evidence points to a 7.6–7.3 Ma age (latest Tortonian) for the type specimen of *N. miocaena*.

**Comments:** Fossil material from the Funakawa Formation (upper Miocene) of Akita, Japan (*Oishi & Hasegawa, 1994*) and isolated tympanic material from the Tatsunokuchi and Na-arai formations (Lower Pliocene) of northeastern Honshu, Japan have been tentatively assigned to this taxon (*Deméré, Berta & McGowen, 2005*). We have not viewed this material and therefore cannot comment on its affinity.

## Description
### Cranium (Figs. 2–4 and 7; Figs. S1–S2)
*Premaxilla.* In dorsal view, the premaxilla is exposed along the mesorostral groove, then narrows posteriorly. The left premaxilla is fragmented into a sharp point anteriorly with

**Table 1  Ten characters of the periotic that broadly distinguish *Balaenoptera* and *Megaptera*, and how *Norrisanima* compares to these genera (Y/N).**

| # | Character | B | M | N | View |
|---|-----------|---|---|---|------|
| 1 | Aperture of CN ducts deeply recessed in PC | N | Y | Y | M, D |
| 2 | Aperture of CN ducts erupt medial; not deflected dorsally | N | Y | Y | M, D |
| 3 | Dorsoventrally flat and mediolaterally elongate PC | N | Y | Y | V |
| 4 | Short and robust AP relative to size of PC | N | Y | Y | V, D |
| 5 | Presence of lateral crest on the ventral surface of AP | N | Y | N | V |
| 6 | Sharply pointed triangular flange | N | Y | N | V, D |
| 7 | Posteriorly deflecting triangular flange | N | Y | N | V, D |
| 8 | Concave medial margin of PC | N | Y | Y | V, D |
| 9 | Transverse ridge on ventral surface of PC | N | Y | N | V |
| 10 | Deep invagination of fenestra ovalis | N | Y | Y | V |

**Notes.**
Column abbreviations: B, Balaenoptera; M, Megaptera; N, Norrisanima.
Character abbreviations: PC, pars cochlearis; CN, cranial nerves; AP, anterior process.
View abbreviations: M, medial; D, dorsal; V, ventral.

a medial deflection so that its medial margin extends posterior to its lateral margin, terminating at about the level of the anterior margin of the nasal. The ascending process of the premaxilla is incomplete where it abuts the left nasal, leaving an open suture. We suspect that, in life, the ascending process of the maxilla likely would have abutted or overlapped the ascending process of the premaxilla near the nasal. Finally, the premaxilla is situated dorsally above the maxilla, so the rostrum slopes ventrally toward the lateral margin.

*Maxilla.* The right maxilla is damaged and almost entirely lost; the anterior section of the left maxilla is damaged and lost. The posterior portion of the ascending process of the left maxilla appears to be lost as well, revealing a slight depression lateral to the left nasal. In dorsal view, both the medial margin of the maxilla is parallel with the sagittal plane, while the lateral margin of the maxilla diverges posterolaterally. The medial margin of the maxilla has a slight concavity, whereas the preserved lateral margin appears straight. Overall, the maxilla broadens posteriorly where it transitions to the rest of the cranium. In the posteromedial corner of the maxilla broadens with an ascending process of unknown shape, although the process does extend medially towards the premaxilla and the parietal—the degree of overlap or interdigitation cannot be distinguished based on the preservation. Similarly, we cannot ascertain, with any degree of confidence, the articulation and sutural configuration of the ascending process of the maxilla and the anterior extent of the parietal at this part of the vertex, where the overall topography curves ventrolaterally from dorsal height of the nasals. At the lateral margin, the maxilla lies ventral to the frontal creating a shallow antorbital notch. In anterior view, the maxilla is elevated dorsally at the medial margin where it contacts the premaxilla and descends ventrally at the lateral margin. The poor preservation of the maxilla prevents determination of any dorsal infraorbital foramina.

In ventral view, the lateral margin of the maxilla is parallel with the sagittal plane until it deflects laterally, slightly anterior to the anterior margins of the frontals. The medial margin

of the maxilla is damaged, but can clearly be seen underlying the vomer. The posterior margin of the maxilla is also fragmented, making it difficult to discriminate an infraorbital plate. Medially, the posterior margin underlies the palatine. The posteromedial corner of the maxilla is poorly preserved. Clear foramina and palatal sulci are present along the whole ventral surface of the maxilla on the rostrum, likely representing the presence of the neurovasculature to support baleen in life. Posteriorly, the foramina and palatal sulci angle posterolaterally, and angle increasingly lateral moving anteriorly. A triangle-shaped trough lies at the posterolateral corner of the maxilla, anteroventral to the preorbital process of frontal (this feature is visible in other extant balaenopteroids). The two maxillae would likely have contacted each other at the midline (possibly with the anterior portion of the vomer visible near the mesorostral groove), but we cannot be certain because of the missing portion of the right maxilla.

*Nasal.* Overall the nasal is short and rectangular, being slightly longer anteroposteriorly than transversely wide. In dorsal view, the left nasal is complete and in its original articulation, but the ascending process of the left premaxilla is missing, leaving a gap between the ascending process of the maxilla and the left nasal. The posterior margin of the nasal abuts the anterior margin of the supraoccipital. The anterior margin curves anteriorly at the lateral corner. An anterolateral projection parallels the premaxilla and creates the posterolateral border of the mesorostral groove. Together, the anterior margins of both nasals overall slope anteroventrally, descending into the mesorostral groove.

*Palatine.* In ventral view, the anterior portion of the palatine is fragmented and it was difficult to determine a clear anterior suture with the maxilla. The lateral margin of the palatine wraps dorsally, and forms the medial margin of the frontal. It is unclear if the posterior margin of the palatine contacts the pterygoid. The palatine slopes ventrally toward the midline where it underlies the vomerine crest.

*Vomer.* In ventral view, the vomerine crest appears below the palatine at the level of the distal opening of the orbital canal. It rises ventrally as it progresses posteriorly and extends under the basisphenoid to the level between the basioccipital crests. At its visible anterior end in the basicranium, the vomer is transversely swollen and narrows posteriorly to form a rounded ridge. The surfaces of the vomer that line the internal choanae are damaged anteriorly, but the shape is clear, with anterior portions that are wide relative to the posterior end, and terminate near the pterygoid-basioccipital suture, anteromedial to the bulbous portion of the basioccipital crests, which form ventrally deflected troughs between each basioccipital crest and the vomerine crest. Posteriorly, the nasal plate of the vomer forms the floor of the basisphenoid.

*Frontal.* In dorsal view, the frontal is broadly rectangular with the lateral and medial margins shorter than the anterior and posterior margins. The anterior margin of the frontal appears to abut the maxilla in a broadly transverse orientation, although the exact contact between the two elements is obsure because of poor preservation. Along the lateral margin, the contact between the lateral process of the maxilla and the front is clear,

suggesting a similar transverse orientation as in other balaenopteroids. In a lateral view the frontal shows a patent but shallow depression of maxillofrontal suture that *Deméré, Berta & McGowen (2005)* termed an "incident balaenopterid 'pocket'," although this structure in other rorquals is largely a construction of the maxilla rather than the frontal. The poor preservation again precludes speculation about its original morphology. In oblique dorsolateral view, the medial margin of the frontal may contact the parietal just before the parietal rises to the vertex, excluding the frontal's participation in the vertex. The true sutural contacts of this relationship are unclear, and we leave the interpretation of this part of the vertex open to further work.

The preorbital process of the frontal is directly posterior to the lateral process of the maxilla. The postorbital process of the frontal extends slightly further laterally relative to the anteroposterior leve of the preorbital process. The medial margin of the frontal near the temporal wall is dorsally elevated so that the lateral margin slopes ventrally. Posteriorly, the postorbital ridge of the frontal is robust and dorsoventrally thickened. Beginning with the contact between the postorbital process and the zygomatic process of the squamosal, a blunt ridge on the dorsal surface of the frontal extends anteromedially to about the level of the antereoposterior middle of the orbit. In life, this ridge likely demarcated the lateral margin for the surface of origin for the temporalis muscle.

In ventral view, the anterior margin of the frontal appears to have a transverse suture with the maxilla overriding it, although poor preservation makes it difficult to ascertain the extent of infraorbital plate of the maxilla overlying the maxillofrontal suture. Medially, the ventral surface of the frontal borders the palatine. The optic canal is deeply concave with posterior and anterior margins (medially) that are dramatically curved ventrally, with the two edges almost touching ventromedially. The preorbital process points anteroventrally, but does not underlie the maxilla. The postorbital process contacts the zygomatic process of the squamosal. No jugal or lacrimal are preserved.

*Parietal and alisphenoid.* In dorsal view, near the vertex, the parietal appears to be exposed at the vertex ventralolateral to the supraoccipital, and possibly in confluence with the medioposterior end of the ascending process of the maxilla. Poor preservation prevents a specific discrimination of the parietal in this region. The presence of an interparietal cannot be determined. Posterior of the level of the vertex, the lateral exposure of the parietal broadens below the overhanging nuchal crest and dorsomedial to the frontal; the parietal appears patent both in the vertical wall descending from the nuchal crest to the frontal, and spreads, in fan shape orientation, along the posteromedial corner of the supraorbital process of the joining with the vertical surface of the same element at nearly 90° angle. The surface of the parietal in the temporal wall is somewhat obscured by poor preservation, but basic sutures can be identified. In lateral view of the wall, the parietal narrows posteriorly, pinching past the posterior extent of the frontal then dorsoventrally expands greatly in the temporal wall where it forms a sigmoidal suture with the squamosal, at the posterior margin of the temporal fossa.

The alisphenoid is visible in the temporal wall vental to the posteromedial corner of the postorbital ridge of the supraorbital process of the frontal. In the temporal wall,

the alisphenoid appears as a trapezoidal window, dorsal to the pterygoid; the squamosal contacts the posterior margin while the parietal contacts the alisphenoid's entire dorsal margin.

*Supraoccipital.* The sutures between the supraoccipital, exoccipital, and basioccipital are tightly ankylosed. In dorsal view, the general shape of the supraoccipital is neither triangular nor circular, but trapezoidal with a lightly squared-off anterior margin. The lateral margins diverge slightly laterally in the posterior two thirds. The anterior of supraoccipital shield passes anterior of the levels of the frontal and parietal, abutting the nasals anteriorly at the vertex. It is relatively flat topographically, with no sagittal crest or obvious foramina or sulci. The nuchal crest comprises the lateral border of the supraoccipital at the supraoccipital-squamosal suture where it rises anteromedially as the continuation of the mastoid crest of the squamosal.

*Exoccipital.* In ventral view, there is a gentle posterolateral deflection of the exoccipital. In lateral view, the paroccipital process is dorsoventrally aligned, flat (i.e., not tilted), and thickened anteroposteriorly. The posterior surface of the paraoccipital reaches posteriorly the level of the occipital condyles. In posterior view, the occipital condyles are reniform in shape and transversely broader ventrally than dorsally; dorsal condyloid foramina are present on both sides. The occipital condyles are large relative to the basiocciptal crests and lay nearly in a dorsoventral plane, with lateral margins strongly convex and medial margins straight. There is a narrow ventral intercondylar notch and a broad dorsal intercondylar notch.

*Basioccipital.* In ventral view, the anterior margin of the basioccipital may be slightly partially obscured by the nasal plate of the vomer. The lateral margins of the basioccipital form the medial margins pterygoid sinus fossa, while the ventral surface of the basioccipital is flat between the basioccipital crests. The crests are massive and rounded (i.e., bulbous). The bone overlying the jugular notch, which is deflected laterally at a 45° angle from the midline, is robust and nearly forms an arch.

*Squamosal.* The squamosal is robust medially at the position of the mastoid crest where it abuts the supraoccipital and exoccipital, and pinches and slants anteroposteriorly as it extends anterolaterally becoming the zygomatic process. In dorsal view, the anterolateral margin of the squamosal visibly forms the posterior margin of the temporal fossa. Medially, this line moves into the temporal wall. In ventral view, the squamorsal borders the pterygoid to form the lateral border of the pterygoid sinus fossa, along the ventral margin of the temporal fossa.

In ventral view, the squamosal has a laterally expansive and deep glenoid fossa, with a large and slightly bulbous postglenoid process. The postglenoid process extends more ventrally than posteriorly, but has an anterior hook at the lateral margin that slightly encloses the glenoid fossa laterally and posteriorly. In lateral view, the anterior margin of the squamosal is more dorsoventrally oriented than anteroposteriorly. The postglenoid process is the most ventral portion of the cranium far below the ventral plane of the

basioccipital crests. In posterior view, the nuchal crest becomes the mastoid crest at about the level of the dorsal termination of the occipital condyles.

The zygomatic process is anteroposteriorly short but dorsoventrally robust and contacts the postorbital process of the frontal. The zygomatic process is taller dorsoventrally than it is transversely wide or anteroposteriorly long. Its overall axis has an anterolateral deflection and is not straight anteroposteriorly. The ventral anterior portion of the right zygomatic process is damaged, but the reconstruction from the early 20th century preserves the overall relationship and distance to the postorbital process of the frontal.

*Pterygoid.* In ventral view, the pterygoid sinus fossa is a large and deep renal-shaped cavity that tapers slightly posteriorly. The roof of the pterygoid sinus fossa is overlain and enclosed posterodorsally by the squamosal. The medial surface of the pterygoid forms the lateral surface of the internal nares; the pterygoid hamulus is missing, likely from a break. The lateral surface of the pterygoid rises from the pterygoid fossa and contributes to the temporal wall.

### Tympanic Bulla (Fig. 5; Fig. S4)

In dorsal view, the tympanic bulla has an overall rectangular shape. It has a slightly pear-shaped medial margin resulting from a shallow median furrow. The posterior edge has rounded prominences (the medial prominence is transversely broader) separated by a shallow interprominental notch. The medial posterior prominence has a rounded transition to the medial margin of the bulla, whereas the lateral posterior prominence transitions to the lateral margin at a sharp angle. The anterior margin is relatively flat, and the lateral surface is slightly convex anteriorly, before transitioning to a pronounced lateral deflection at the level of the sigmoid process. As a result, a deep lateral furrow separates the middle portion from the anterior portion. Continuing posteriorly from the sigmoid process, the lateral margin is straight in the sagittal plane.

The eustachian notch is broad and directed medially. The anterior portion of the outer lip is broken so the anterior pedicle is not present. The sigmoid process is poorly preserved and separate from the body of the bulla, but a contact can be inferred. It originates posterior from the level of the lateral furrow and overlays the tympanic cavity with a posterior deflection. The posterior pedicle is not preserved. The involucrum is transversely broad posteriorly and narrows anteriorly. Posteriorly it is smooth, but there are anterior transverse creases emanating from the eustachian notch.

In ventral view, the surface of the tympanic is relatively smooth throughout the body of the bulla, except the medial margin where rugose pitting is visible. Although there is some damage to the outer lip, the anterolateral corner is inflated, creating a distinct lobe bound posteriorly by the lateral furrow. There is no anterolateral shelf.

In lateral view, the bulla is somewhat ovoid. The anterior margin is dorsoventrally aligned and transitions gently into the smooth convex ventral margin. The ventral margin ends abruptly at the lateral posterior prominence and angles sharply anterodorsally as it becomes the posterior margin. In lateral view the posterior deflection of the sigmoid process is readily apparent, such that it extends posterior to the conical process. An elongate

projection of the sigmoid process is preserved, but not modeled because it is a separate fragment, free from articulation with the rest of the bulla. Anterior to the sigmoid process, the conical process is preserved as a blunted peak. Damage to the lateral surface prevents interpretation of the mallear ridge and sigmoidal cleft.

In medial view, the involucrum is massively globular posteriorly and narrows anteriorly. The involucral ridge is shallow, and oriented anteroposteriorly with a very slight ventral convexity. This ridge is nearly parallel to the main ridge and separated from it by a band of rugose and deeply pitted bone. The medial posterior prominence is bulbous and smooth with a dorsoventrally straight posterior margin. The median furrow has a slight anterior bulge.

### Periotic (*Figs. 6* and *8*; *Fig. S3*)

Of the two periotics preserved in the type specimen, the left periotic is better preserved and provides the basis for the following description. In dorsal view, the periotic is roughly L-shaped, and consists of a triangular anterior process and two medial projections of the pars cochlearis. The posterolateral angle of the periotic is the triangular flange of the lateral tuberosity; the posterior processes of both periotics are fused with the posterior processes of the tympanic bullae, and they remain preserved *in situ* with the cranium. The lateral border of the anterior process is rounded anteriorly, while the medial edge is dorsoventrally oriented. The most lateral portion of the periotic is an inflated lateral margin of the anterior process. The posterior edge of the pars cochlearis exhibits a shallow concavity, bordered laterally by the level of the caudal tympanic process. The external anterior margin of the anteriormost of the two projections of the medial extension of the pars cochlearis is straight and medially oriented. The internal margin of this projection is rugose. The posterior projections follows the same aspect but it is not as long in the medial direction.

In ventral view, the anterolateral sulcus is present and follows an anteroposterior direction offset laterally from the anterior tip of the anterior process. The medial margin of the anterior process is sinusoidal and terminates posteriorly near the level of the hiatus fallopii. The facial sulcus wraps posterolaterally around the base of the smooth surface of the cochlea. The posteromedial termination of the facial sulcus is ventral to, and obscures, the opening of the fenestra ovalis near the triangular flange of the lateral tuberosity. The opening of the fenestra ovalis is occupied by a relatively large stapes in articulation.

In medial view, the periotic appears as two distinct sections, the anterior process (which is roughly conical is shape) and the pars cochlearis (which appears roughly spheroid with a posterior projection off the posterodorsal corner—the lateral ridges of the triangular flange of the lateral tuberosity). The internal auditory meatus is located roughly between the two major projections extending medially from the pars cochlearis. Many of the features of the internal auditory meatus and the area surrounding it are indicative of an ontogenetically mature individual, especially relative to the work of *Ekdale, Berta & Deméré (2011)* on ontogenetic changes in the shape and depth of the internal auditory meatus in *M. novaeangliae*. These traits include: (1) a nearly circular shared aperture for both the facial canal (CNVII), and (2) the vestibulocochlear aqueduct (CNVIII) separated by a deeply

recessed crista transversa. The crista transversa runs relatively straight anteroposteriorly with the facial canal on the ventral side and the vestibulocochlear canal on the dorsal side. Posterior to the internal auditory meatus is the recessed and dorsoventrally elliptical opening of the perilymphatic and endolymphatic foramina.

In lateral view, the periotic is flat along the dorsal margin aside from a concavity near the posterior half of the anterior process. The ventral margin of the anterior process is also relatively flat and terminates posteriorly at the cochlea. The cochlea protrudes ventrally quite abruptly before curving posteriorly and then sloping back toward the dorsoposterior corner.

## DISCUSSION

### *Norrisanima* compared with crown balaenopteroids

In recent phylogenetic analysis of mysticetes using morphological and molecular data, *Marx & Fordyce (2015)* and *Slater, Goldbogen & Pyenson (2017)* showed *Norrisanima* represented a lineage positioned well outside of crown Balaenopteroidea, unrelated to species in the extant genus *Balaenoptera*, *Eschrichtius*, and not sister to *M. novaeangliae*. The morphological partition of *Slater, Goldbogen & Pyenson (2017)*'s analysis (which was based data from *Marx & Fordyce, 2015*) shows seven synapomorphies that diagnose crown Balaenopteroidea: (1) a straight posterior border of the supraorbital process in dorsal view; (2) a short postorbital process that does not markedly project in any direction; (3) an optic canal that in ventral view is enclosed medially by bony laminae; (4) a well-developed and thickened postorbital ridge along the medial portion of the optic canal; (5) flattened dorsal surface of the nasal bones; (6) inflated posterior corner of the pars cochlearis (medial to the fenestra rotunda) that extends posteriorly beyond the fenestra rotunda; and (7) absent or indistinct medial lobe of the tympanic bulla. *Norrisanima* exhibits only one of these seven synapomorphies: the flattened nasal bones. This trait is a marked departure from the rounded condition of some stem mysticetes; and some crown mysticetes (and even *Balaenoptera* spp.) that exhibit nasals with a peak or crest extending to the midline. Although there is no right nasal in the holotype of *Norrisanima*, the left nasal is clearly flat and only curves ventrally at the anterior-most margin as it dives into the mesorostral groove.

The other six traits separate *Norrisanima* from crown Balaenopteroidea. Instead of a straight posterior edge of the supraorbital process, *Norrisanima* has a slightly concave margin. Like crown Balaenopteroidea, *Norrisanima* does have a short postorbital process, but it is deflected posteriorly instead of laterally as in crown balaenopteroids. The medial portion of the optic canal is open ventrally in *Norrisanima*, although anterior and posterior margins of the optic canal do extent ventrally toward the medial end of the canal, and are almost touching. This state in *Norrisanima* appears to be somewhat intermediate between stem and crown balaenopteroids, which have closed optic canals medially. *Norrisanima* clearly has a postorbital ridge, but it is not as thick as modern rorquals, nor does it displace the optic canal from the posterior border of the supraorbital process as in extant balaenopteroids.

Broadly, the entire vertex of *Norrisanima* is very reminiscent of crown balaenopteroids, especially large species in the genus *Balaenoptera*, such as *Balaenoptera musculus* (*Linnaeus, 1758*) and *Balaenoptera physalus* (Fig. 6). In both of these latter species, the nasals and ascending processes of the premaxillae meet the anterior margin of the supraoccipital shield in a transverse line that is nearly rectilinear for all the element terminations involved (unlike, for example, the posterior pinching of the nasals in *B. acutorostrata Lacépède, 1804*). However, in *Norrisanima*, the position of this configuration of the vertex relative to the level of orbit is most similar to *B. acutorostrata*, nearly at the midway level between the pre- and postorbital processes of the frontal. The outline of the nasal in dorsal view in *Norrisanima*, in particular, shares broad rectangular features with *Eubalaena australis* (*Desmoulins, 1822*) and an anterolateral spur similar to the one found in some specimens of *B. edeni Anderson, 1878* (see *Omura et al., 1981*).

The tympanoperiotics of *Norrisanima* show two traits that are not shared with crown balaenopteroids, although the differences are subtle (Fig. 7). For example, the pars cochlearis is less globular and inflated in *Norrisanima* than in crown balaenopteroids (*Ekdale, Berta & Deméré, 2011*). Also, medial to the fenestra rotunda, *Norrisanima* exhibits a raised, inflated surface reminiscent, in miniature, of the involcrum on the tympanic bulla; this inflated surface is absent in members of the genus *Balaenoptera* (*Ekdale, Berta & Deméré, 2011*). Also, crown balaenopteroids do not have a medial lobe on their tympanic bulla, or this medial lobe is indistinct. *Norrisanima* has a medial lobe on the bulla that is equal in size to the lateral lobe.

Because of the taxonomic legacy of this specimen, we further compared *Norrisanima* with *Megaptera* and *Balaenoptera* spp., focusing on the periotics, which possess a large number of diagnostic cetacean traits (*Ekdale, Berta & Deméré, 2011*). This process was performed in two stages: first, we compared *Megaptera* and *Balaenoptera* to develop a list of traits in which the two generally differ; then we compared *Norrisanima* to each genus within the context of these traits. These traits are not meant to be diagnostic nor exhaustive, and are merely heuristic; *Norrisanima* is formally diagnosed above according to previous phylogenetic analyses, not using this set of traits.

In comparisons between extant *Megaptera* and *Balaenoptera*, we found ten characters that broadly distinguish the two genera (listed in Table 1). *Megaptera* periotics broadly differ from other *Balaenoptera* periotics in ten features, including: (1) the apertures of the cranial nerve ducts erupt in a deep recess in the medial margin of the pars cochlearis; (2) the apertures of the cranial nerve ducts open medially, without a dorsal deflection found in *Balaenoptera*; (3) a dorsoventrally flattened and mediolaterally elongate pars cochlearis; (4) a short and robust anterior process relative to the size of the pars cochlearis; (5) a lateral crest on the ventral surface of the anterior process; (6) a sharply pointed triangular flange of the lateral tuberosity; (7) a posteriorly deflected triangular flange; (8) a concave medial margin of the pars cochlearis (in dorsal and ventral view); (9) a transverse ridge on the ventral surface of the pars cochlearis; and (10) a deep invagination of the fenestra ovalis that almost completely obscures the stapes in ventral view.

*Norrisanima* shares six traits with *Megaptera* and four with *Balaenoptera* (Table 1). The six traits shared with *Megaptera* include: (1) the apertures of the cranial nerve ducts erupt

in a deep recess in the medial margin of the pars cochlearis; (2) the apertures of the cranial nerve ducts open medial, without a dorsal deflection; (3) a dorsoventrally flattening and mediolaterally elongation of the pars cochlearis; (4) a short and robust anterior process relative to the size of the pars cochlearis; (5) a concave medial margin of the pars cochlearis (in dorsal and ventral view); and (6) a deep invagination of the fenestra ovalis that almost completely obscures the stapes. The four traits that *Norrisanima* shares with *Balaenoptera* include: (1) the lack of a lateral crest on the ventral surface of the anterior process; (2) a rounded pointed triangular flange of the lateral tuberosity; (3) a laterally deflected triangular flange; and (4) the lack of a transverse ridge on the ventral surface of the pars cochlearis.

## Comparisons with other fossil mysticetes

As a stem balaenopteroid, the holotype specimen of *Norrisanima* shares some similarities with other stem balaenopteroids and fossil mysticetes of similar age, including '*Balaenoptera*' *siberi* Pilleri, 1989 (also see Pilleri, 1990), '*Megaptera*' *hubachi* Dathe, 1983 and *Incakujira anillodefuego* Marx & Kohno, 2016. The proximal end of the rostrum of *Norrisanima* is similar to all of these taxa, although the incompleteness of the type specimen makes comparisons difficult; indeed few fossil balaenopteroid taxa (either crown or stem) preserve the entire rostral margin intact. The lateral process of the maxilla in *Norrisanima* is not a perpendicular deflection as in *Protororqualus cuvieri* (Bisconti, 2007b), but about 120° from the midline, more like *Incakujira*, '*B.*' *siberi*, and *Plesiobalaenoptera quarantellii* Bisconti, 2010a.

The overall shape of the nasal in *Norrisanima* is somewhat similar to '*B.*' *siberi*, '*M.*' *hubachi* Bisconti, 2010b and *Protororqualus cuvieri* Bisconti, 2007b, but the laterally even and rectilinear nasal outlines of *Norrisanima* are unique; all other stem and crown balaenopteroids have nasals that taper posteriorly. On the dorsal surface of the nasal, *Norrisanima* possesses an anterolateral flange similar, but longer, than those present in *B. edeni* (Omura et al., 1981). In dorsal view, *Archaebalaenoptera castriarquati* Bisconti, 2007a has an anterolateral corner of the nasal that exceeds the anterior level of the anteromedial corner, but there is no flange as in *Norrisanima*. Although somewhat incomplete, the remains of what appears to be a relatively thick premaxilla near the anterior termination of the nasals in *Norrisanima* is similar to that found in small to mid-sized *Balaenoptera* spp., such as *B. acutorostrata* and *B. edeni*, and less like the thinner terminations in *Archaebalaenoptera*, *Incakujira*, *Parabalaenoptera*, *Protororqualus*, '*M.*' *hubachi*, and *Nehalaennia devossi* Bisconti, Munsterman & Post, 2019.

The medial surface of the supraorbital process of the frontal does not slope to the vertex as in true cetotheriids, such as *Joumocetus shimizui* Kimura & Hasegawa, 2010 although it is not as sharply tabular, where the vertex is stepped above the level of the frontal, as in '*B.*' *siberi*, *Incakujira*, and *Balaenoptera* spp. Generally, the vertex in *Norrisanima* is transversely wide (relative to the length of the nasals) compared to *Archaebalaenoptera*, *Incakujira*, *Parabalaenoptera*, *Protororqualus*, and even '*M.*' *hubachi*. The dorsal junction of the maxilla and frontal that forms the so-called "balaenopterid pocket" (Deméré, Berta & McGowen, 2005) is lightly visible in *Norrisanima*, apparently to the same degree as in '*M.*' *hubachi*,

but certainly not as strongly delineated as in *Archaebalaenoptera, 'B.' siberi, Incakujira, Protororqualus, Nehalaennia,* and living *Balaenoptera* and *Megaptera.* In dorsal view, the postorbital process of the frontal in *Norrisanima* is sharply angular and notably overlays the zygomatic process of the squamosal; this combination of features is clear in *Incakujira,* slightly overlapping in '*M.' hubachi,* but notably absent in other stem balaenopteroids with complete supraorbital processes of the frontals, such as *Parabalaenoptera.*

In dorsal view, *Norrisanima* shares the overlap of the parietal on to the posteromedial corner of the supraorbital process of the frontal with other stem balaenopteroids such as *B. bertae, Protororqualus, Nehalaennia,* and to a small degree *Archaebalaenoptera,* but not *Incakujira,* and other crown balaenopteroids. It appears that ''*Balaenoptera*'' *ryani Hanna & McLellan, 1924* also possessed such overlap (as in *Norrisanima*); as *Deméré, Berta & McGowen (2005)* pointed out, the type and only specimen of this taxon requires a redescription and likely a new generic name.

In *Norrisanima,* the anterior margin of the supraoccipital shield is essentially at the level of the preorbital process, in dorsal view, which is broadly similar to living *Megaptera* and *B. acutorostrata,* but unlike all other living *Balaenoptera* spp. The position of this margin in *Norrisanima* is anterior to most other stem balaenopteroids, where it is shifted more posteriorly, as in '*M.' hubachi, Parabalaenoptera, 'B.' siberi, Nehalaennia, Balaenoptera bertae Boessenecker, 2012* and *Incakujira.* Also, the shape of this margin in *Norrisanima* is broadly semi-lunar with notable angularity resulting in a slight rectangular profile, not sharply acute as in *Nehalaennia, Incakujira, Parabalaenoptera,* and *B. bertae,* or with the irregular lobate margin as in '*M.' hubachi.* The lateral margin of the supraoccipital, extending to the nuchal crests, overhangs the posterior margin of the temporal wall in *Norrisanima,* as it does in most other fossil balaenopteroids, but not *Archaebalaenoptera.* In shape and relative position to the frontal, the supraoccipital of *Norrisanima* shares little with *Archaebalaenoptera* and *Protororqualus.* No interparietal is visible in the dorsal vertex of *Norrisanima,* although it is possible that part of the parietals are exposed along the lateral margins of the vertex; it is difficult to ascertain this feature because of poor preservation.

The length and lateral deflection of the zygomatic processes of the squamosal in *Norrisanima* resembles that of living *Balaenoptera* spp., and especially *Megaptera.* Like in many fossil balaenopteroids, the postglenoid process and the posterolaterally facing suprameatal fossa are not visible in dorsal view of the basicranium of *Norrisanima.* The ratio of the width of the paroccipitals, relative to the bizygomatic width, is more like that in *Incakujira, Parabalaenoptera* and '*M.' hubachi* than that in living *Balaenoptera* and *Megaptera.* The entire ventral side of the basicranium in *Norrisanima* is broadly proportioned like living *Balaenoptera* and *Megaptera,* matching the distance between the glenoid fossa to the pterygoid sinuses, and the anteroposterior length across the observable space formed by the temporal fossa in this view. Although the type specimen of *Norrisanima* lacks a mandible, the glenoid fossa is broadly similar to extant rorquals, suggesting some basis for inferring lunge-feeding features in this taxon, although there are important soft tissue features that lack osteological correlates to strengthen this argument.

Lastly, we calculated an estimated total length of *Norrisanima* at 12.49 m (using *Pyenson & Sponberg* (*2011*)'s reconstruction equation for stem balaenopteroids). This body size

matches that extant gray and sei whales, and is the largest stem balaenopteroid known (see *Slater, Goldbogen & Pyenson, 2017*). Interestingly, *Norrisanima* is essentially the same size as the late Miocene to early Pliocene age *Eubalaena shinshuensis* (12.46 m; *Slater, Goldbogen & Pyenson, 2017*), and together these two fossil taxa are larger than any stem mysticete and any crown mysticete outside of crown Balaenopteroidea and crown Balaenidae.

### Late Miocene marine mammal assemblages from California and phylogenetic divergence times

*Norrisanima* was collected from late Miocene age diatomite sequences of southern California that have also yielded a variety of large marine vertebrates, including type specimens of the pinnipeds *Pithanotaria starri Kellogg, 1925* and *Imagotaria downsi Mitchell, 1968*, and a variety of other seabird and fossil vertebrate taxa. The Tortonian age constraints of *Norrisanima* (7.6–7.3 Ma) potentially also apply to *Pithanotaria* and *Imagotaria*, which were collected from likely coeval units of the diatomite sequences with the type locality of *Norrisanima*, which collectively provide narrower stratigraphic intervals on divergence dates for clades related to these taxa. For example, *Pithanotaria* is a crown otariid according to recent analyses (*Velez-Juarbe, 2017*), and as the oldest crown lineage, its age constrains the minimum divergence time for this clade.

*Norrisanima*'s position outside of crown Balaenopteroidea limits its use to constrain the divergence date for crown Balaenopteroidea, although as a late Miocene (Tortonian age) balaenopteroid, *Norrisanima* is notably older than most other Messinian and Pliocene age fossil balaenopteroids, including all extant and fossil eschrichtiids (see *Deméré, Berta & McGowen, 2005*; *Marx & Fordyce, 2015*; *Slater, Goldbogen & Pyenson, 2017*). The estimated divergence date for crown Balaenopteroidea is likely older than the late Miocene; *McGowen, Spaulding & Gatesy* (*2009*: Table 2, Node 7) provided a comprehensive overview of molecular divergence dates for cetaceans, and calculated a mean Middle Miocene age (13.80 Ma) for the divergence of crown Balaenopteroidea, and largely argued for a 12–10 Ma (later Middle to early Late Miocene) origin timeframe. *Geisler et al. (2011)* proposed *Norrisanima* as a preferred fossil calibration point for the minimum divergence age of Plicogulae (i.e., the crown clade formed by *Caperea* + *Balaenoptera*). *Geisler et al. (2011)* indicated a 11.6–7.2 Ma age range for *Norrisanima*, with a 7.2 Ma date as a minimum age. We propose that this range be narrowed to 7.6–7.3 Ma and the refined age of this taxon would make it more useful for tip-dating calibrations of Plicogulae (see *Ronquist et al., 2012*; *Parham et al., 2012*).

## CONCLUSIONS

Phylogenetically accurate taxon names for stem and crown taxa are essential for examining the course of evolution giving rise to modern biodiversity. Baleen whales (Mysticeti) represent organismal maxima for many biological traits, but their evolutionary history, including the description of their fossil record, remains a work in progress. *Norrisanima miocaena* is an extinct species that lacks balaenopteroid synapomorphies as well as all of the autapomorphies of extant *M. novaeangliae*. Our observations accompany the current phylogenetic framework resulting from broad molecular and morphological datasets,

which place *N. miocaena* outside of crown Balaenopteroidea, and unrelated to extant humpback whales. By redescribing this fossil species and giving it a new generic name, we were also able to more narrowly constrain its Tortonian age, which in turn provides constraints for fossil calibrations of several divergence times in mysticete and pinniped history.

**Institutional abbreviations**

**USNM**    Departments of Paleobiology (holotype) and Vertebrate Zoology (Division of Mammals for comparative specimens), National Museum of Natural History, Smithsonian Institution, Washington, District of Columbia, USA.

## ACKNOWLEDGEMENTS

We thank M Dattoria, J Conrad, V Rossi, A Metallo, and the Smithsonian Institution's Digitization Program Office 3D Lab for training, technical support, and access to equipment. We thank J Hinton and S Sholts of the Smithsonian Institution Bio-Imaging Research Center in the Department of Anthropology at the National Museum of Natural History for assistance CT scanning the tympanoperiotic material. We also thank D Lunde, J Ososky, and MR McGowen for access to the collections in the Division of Mammals in the Department of Vertebrate Zoology at NMNH. MSL thanks the Smithsonian Office of Fellowships and Internships—especially A Lemon and A Capobianco—for their assistance during his James Smithson and Secretary's Distinguished Research Postdoctoral Fellowship. NDP thanks the Basis Foundation and its Remington Kellogg Fund for support and JF Parham and J Velez-Juarbe for comments on the manuscript. We express great thanks to Katherine D. Klim for help proofreading this manuscript. Finally, we thank E Coombs, E Noirault, A Goswami, and R Sabin for access to 3D digital models from the Natural History Museum London.

### Funding

This work was supported by the Smithsonian Institution's Remington Kellogg Fund and the Basis Foundation. Matthew S. Leslie was supported by the Smithsonian Institution's G. Wayne Clough fellowship (part of the James Smithson fellowship program) and the Smithsonian Institution's Secretary's Distinguished Postdoctoral Fellowship. The funders had no role in study design, data collection and analysis, decision to publish, or preparation of the manuscript.

### Grant Disclosures

The following grant information was disclosed by the authors:
Smithsonian Institution's Remington Kellogg Fund and the Basis Foundation.
Smithsonian Institution's G. Wayne Clough fellowship (part of the James Smithson fellowship program).
Smithsonian Institution's Secretary's Distinguished Postdoctoral Fellowship.

## Competing Interests

Nicholas D. Pyenson is an Academic Editor for PeerJ.

## Author Contributions

- Matthew S. Leslie, Carlos Mauricio Peredo and Nicholas D. Pyenson conceived and designed the experiments, performed the experiments, analyzed the data, contributed reagents/materials/analysis tools, prepared figures and/or tables, authored or reviewed drafts of the paper, approved the final draft.

## Data Availability

All 3D data are available on Zenodo: Leslie, Matthew S., Peredo, Carlos M., & Pyenson Nicholas D. (2019). Dataset for Leslie, Peredo & Pyenson 2019 [Data set]. Zenodo. http://doi.org/10.5281/zenodo.3431395.

Periotic specimens are available at the Smithsonian Institution's National Museum of Natural History. Holotype material for *N. miocaena* (USNM 10300) are stored in the Department of Paleobiology. Specimens used for comparisons, including *Megaptera novaeangliae* (USNM 486175), *Balaenoptera borealis* (USNM 504699), *B. physalus* (USNM 237566), and *Balaenoptera bonaerensis* (USNM 504953) are accessioned within the department of Vertebrate Zoology, Division of Mammals.

3D models of several specimens from the Natural History Museum London (NHMUK) were used for comparisons of the cranium include *Caperea marginata* (NHMUK 1876.2.16.1), *Eubalaena australis* (NHMUK 1873.3.3.1), *Balaenoptera edeni* (NHMUK 1920.12.31.1), *Megaptera novaeangliae Borowski, 1781* (NHMUK 792a), *Balaenoptera physalus* (NHMUK 1862.7.18.1), *Balaenoptera acutorostrata* (NHMUK 1965.11.2.1), *Balaenoptera borealis* (NHMUK 1934.5.25.1), and *Balaenoptera musculus* (NHMUK 1892.3.1.1). All NHMUK scans were downloaded from https://doi.org/10.5519/0020467, with the exception of the blue whale which was acquired from https://sketchfab.com/NHM_Imaging (*Sabin et al., 2018*), and the humpback whale, which was made available directly from the NHMUK to the authors.

## New Species Registration

The following information was supplied regarding the registration of a newly described species:

Publication LSID: urn:lsid:zoobank.org:pub:95CFDD42-D8DB-4DC7-BFB3-5B34CCC6508C.

*Norrisanima* nov. gen. LSID: urn:lsid:zoobank.org:act:E777170E-03BC-40AA-A04B-65CE92C956BD.

*Norrisanima miocaena* nov. com. LSID: urn:lsid:zoobank.org:act:E777170E-03BC-40AA-A04B-65CE92C956BD.

## Supplemental Information

Supplemental information for this article can be found online at http://dx.doi.org/10.7717/peerj.7629#supplemental-information.

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
