# Peer review of "Norrisanima miocaena, a new generic name and redescription of a stem balaenopteroid mysticete (Mammalia, Cetacea) from the Miocene of California"

_PeerJ, doi:10.7717/peerj.7629_

## Round 0.1 · original submission · Major Revisions

The specimen described is an important piece in the puzzle of the evolution of modern mysticetes. Its reference to a modern genus has been followed by most authors, but lately a consensus is growing that this is misleading with regard to its actual position and that it deserves its own generic name. The authors are not the first to suggest this, but they are the first to name and formally diagnose the specimen.

The most serious revision I am suggesting follows what the reviewers made very explicit: this manuscript is under-illustrated.

There is a long section describing periotic characters, but few of these are visible in illustrations. Comparative illustrations would be especially useful. You have one, of the dorsal part of the brain case, but other anatomical regions are discussed in detail and not illustrated. While I agree with reviewer 1 in a number of detailed comments, I do not believe a full phylogenetic analysis is necessary before the manuscript can be published.

Reviewer 1 ·

Basic reporting

The 3D model and CT scans are not presented in the supplemental material and cannot be reviewed.

Experimental design

See below.

Validity of the findings

See below.

Additional comments

Balaenopteridae is the largest family of living baleen whales. Its taxonomy is complicated due to morphological diversity and contradictions between the morphological and molecular data which are partly explained by introgressive gene flow (Árnason et al. 2018). 'Megaptera' miocaena is a well known, well preserved specimen which is, possibly, the earliest member of this family, and its re-description using the newest morphological and phylogenetic methods, as well as basic revision of balaenopterid taxonomy at the genus level, would be highly appreciated.

However, the work presented in the manuscript in its current form is far from being complete. The verbal description adds little to the original description by Kellogg (1922). The illustrations show only the surface scans lacking the photographs and the 3D component. The diagnostic traits are not specifically illustrated, and there is no comparative illustrations showing the closely related taxa. The illustrations of the periotic are especially non-informative, and the CT scanned inner structure is not presented, although reported in the Material and Methods section.

The differential diagnosis for the genus is not formulated that is the more important because there is no modern, updated diagnosis for genus Balaenoptera in the light of molecular genetic data. Moreover, while the genus Balaenoptera is pending revision, naming new balaenopterid genera will be debatable. The diagnosis in its present form is based on autopomorphies backing to phylogenetic studies by Marx and Fordyce (2015) and Slater et al. (2017), and no new phylogenetic research has been conducted here. Noteworthy, the apomorphies of the occipital shield are similar to the living Megaptera (lines 680-683), and the same can be said on the shape of the zygomatic process. The specific traits of the periotic bone also are similar either to Balaenoptera spp., or to Megaptera, as admitted by the authors. Thus, the diagnosis should be re-defined as a clear differential diagnosis at the genus level.

No comparison has been provided for 'Balaenoptera' ryani, which is the most closely related taxon for M. miocaena, according to the phylogeny by Marx and Fordyce (2015), and the comparison with B. bertae is rather limited. The comparative drawings showing these taxa are necessary, and taxonomic opinions on them will be helpful.

The comparative part of the manuscript, including the supplementrary material, seems to be incomplete and needs to be finalized. In particular, this concerns the comparison with Incacujira (see the question marks and yellow highlights in the submitted version).

The taxonomic position of 'Megaptera' miocaena have been discussed in detail in two papers with the same conclusion.
As Deméré et al. (2005) reported, "this basal balaenopterid taxon is not a species of Megaptera and, pending additional study, should be placed in a new genus. ... Although originally assigned to Megaptera, the holotype does not possess any of the autapomorphies of the extant humpback whale..."
Marx and Fordyce (2015): "If the nesting of grey whales within crown balaenopterids turned out to be wrong, our analysis would still indicate that grey whales are more closely related to living rorquals than certain balaenopterid stem taxa (e.g. ‘B.’ ryani). Further evidence to support this view comes from the recent description of the bony labyrinth of a range of living and fossil mysticetes, which demonstrated the occurrence of a tympanal recess in eschrichtiids and crown balaenopterids, but not ‘Megaptera’ miocaena [69]. This complements our results, which consistently interpret ‘M.’ miocaena as basal relative to both living rorquals and grey whales (figures 2; electronic supplementary material, figures S2 and S3)."

Therefore, erecting a new genus here is a formal recognition of previous studies and opinions which have become a matter of general consensus among researchers and should be properly respected. At least, the comments by Tom Deméré and Felix Marx will be an important, if not necessary, part of the publication process.

The schematic phylogenies provided in the Figure 1 do not contain the references to the original sources.

Finally, naming a higher rank taxon of Pan Balaenopteroidea is not substantiated, as it is not clearly supported by phylogenetic study. On the contrary, there is a robust definition of Balaenopteroidea as the clade pooling together Balaenoptera, Eschrichtius and the genera which are closer related to them as to Caperea (Steeman, 2007; Marx and Fordyce, 2015), so adding a new rank instead of it is redundant.

Thus, the diagnosis, comparisons and corresponding illustrations should be reworked before the publication, and the previous works on phylogeny and nomenclature should be fully addressed.

·

Basic reporting

The English is clear and easy to follow. I only made a few suggestions to slightly improve some sentences (constructions somewhat unusual, see annotated pdf).
The list of refences is sufficient, but the field background should be presented in a less ambiguous way, clearly stating the input of previous studies on our present knowledge of this taxon (e.g., geological age and phylogenetic relationships, see comments below and in the text).
The paper is well organized, and only a few short parts should be moved elsewhere (e.g., size estimates are provided in the introduction, but should be moved to the results).
All figures are fine and useful. However, see below my comments on the lack of some important views (lateral and posterior) and the probable complementarity between images taken from 3D models and photos. I also made suggestions in the text for the addition of labels in line drawings for diagnostic features.

Experimental design

This paper is about the review of a previously described extinct mysticete species. Part of the results can then be considered as new (description and comparison, with new illustrations), but as mentioned above I would suggest more clearly separating these new results from results taken from previous work (phylogeny, biostratigraphy). This should especially be clearly stated in the abstract and in figure captions.
This is a useful and carefully done revision, but apart from providing a new genus name, the main goals of the work could be more explicitly mentioned in the introduction and abstract.
Methods for the surface scanning are described in detail.

Validity of the findings

Morphological interpretations have been carefully undertaken. In a few places I question the interpretations, but I think all these minor issues should be easily fixed. As mentioned above and below, some anatomical traits could be better supported if additional views and photos (in addition the 3D views) were provided.
Most of the conclusions are well stated, but I think that the part on the divergence of crown Balaenopteroidea should be revised (see comments below and in annotated pdf). Again, this won't need much work, but will have consequences on the content of the abstract.

Additional comments

- Considering that the geological age of the type specimen and the updated phylogenetic relationships were already mentioned in previous works, it would be important to better highlight the main contributions of the present study to our understanding of this interesting taxon (see below the comment on constraints for divergence dates).

- The authors only provide views taken from 3D models as illustrations of the skull. In some cases, these views are undoubtedly better to identify morphological features (for example for ear bones). On the cranium, this is less obvious when looking at Kellogg’s original photos. I made comments in the text concerning regions that would deserve more detailed views, for example the vertex in dorsal view, with interpretive line drawing showing the relationships between nasal, frontal, parietal, and supraoccipital. Also, lateral and posterior views of the skull should be provided as main text figures. if not possible with the 3D model, then photos should be taken. Note that no posterior view was provided in Kellogg's original paper, so this would be especially useful.

- Line 714 : ‘Similarly, Norrisanima’s position outside of crown Balaenopteroidea…’ : This is not similar. For Pithanotaria the taxon is in the crown group, and thus indeed provides a minimum age for the divergence of the crown group. Here you deal with a stem balaenopteroid, so it does not provide a constraint for the crown group (Norrisanima may have existed well before the divergence of crown balaenopteroids, but also well after). I think that you only provide a constraint for the stem Balaenopteroidea. This section should be revised in detail.

- Please check for all taxa the way to provide authorship:
authors, date: for species in their originally described genus
(authors, date): for species that were later referred to another genus

- Figure 1: The caption should preferably be understandable without the main text, so I would suggest adding here at least one or two references for each topology figured here. Especially if no phylogenetic analysis was performed during this study.

- Please see the annotated pdf of the ms for additional comments and suggestions, both in the text and figures.

This is a fine and useful work, and even if the revision may take some time, it will certainly make a good contribution. I am looking forwards to seeing this paper published.

O Lambert

---

## Round 0.2 · Minor Revisions

This manuscript is significantly improved and has not been re-reviewed.I have some very minor remaining suggestions.

Line 80 ‘places’ not place
Line 119 Balaenopteroids misspelled
Line 128 You list as only museum acronym the USNM Division of Mammals. The fossil you describe has a USNM number. Is it part of the Division of Mammals or VP?
Line 173 Joyce et al.
Line 219, do we really need ‘pere et fils’?
Line 625 and 627 optic canals, not optical canals
Line 642, period missing

---

## Round 0.3 · accepted · Accept

I believe that our understanding of the fossil history of baleen whales is attracting a lot of interest, and I am glad that specimens that were collected and described a long time ago, such as the ones in this paper, are restudied and interpreted in a modern context.